# Off-target sequence variations driven by the intrinsic properties of the Cas–sgRNA–DNA complex in genome editing

Celine Kurniawan[1], Takeshi Itoh[1,2]*

**1** Master Program in Global Agriculture Technology and Genomic Science, International College, National Taiwan University, Taipei, Taiwan, **2** Center for Computational and Systems Biology, National Taiwan University, Taipei, Taiwan

* taitoh@ntu.edu.tw

## Abstract

Genome-editing technologies hold significant potential across various biotechnological fields, yet concerns about possible risks, including off-target mutations, remain. To ensure safe and effective application, these unintended mutations must be rigorously examined and minimized. Computational approaches are anticipated to streamline the detection of off-target mutations; however, the performance of current prediction tools is limited, likely owing to insufficient knowledge of off-target mutation characteristics. In this study, we collected experimentally validated off-target mutation data and conducted a large-scale analysis of 177 nonredundant datasets obtained from six studies. We developed a method to assess the statistical significance of sequence pattern similarity and diversity between off-target sites. This method is based on a comparison of ordered relative entropy values for aligned target sequences, and it was compared with two other methods on the basis of Euclidean distance and the Pearson correlation coefficient. The three methods demonstrated clear correlations, indicating their validity. These methods were applied to 238 dataset pairs for the same target site, and it was revealed that off-target sequence patterns were quite similar across different experimental conditions, such as varying cell lines and independent experiments, suggesting that the intrinsic properties of the Cas–sgRNA–DNA complex play a key role in determining cleavage sites. However, newly engineered enzymes and those from different bacterial sources occasionally display unique off-target patterns, indicating the need for comprehensive evaluation of each new enzyme to develop reliable prediction tools. The insights gained from this study are expected to contribute to a better understanding of off-target mutation characteristics and support the development of more accurate computational prediction methods.

**Data availability statement:** All relevant data are within the manuscript and its Supporting Information files.

**Funding:** This work was supported by the Office of Research and Development New Faculty Grant (111L7472), National Taiwan University, and Grant 111-2313-B-002-006-MY3 from the National Science and Technology Council, Republic of China (Taiwan). There was no additional external funding received for this study.

**Competing interests:** The authors have declared that no competing interests exist.

## 1. Introduction

Over the past two decades, advances in genome editing have empowered scientists to make precise modifications to the genomes of various organisms [1]. Among these technologies, RNA-guided CRISPR (clustered regularly interspaced short palindromic repeats)–Cas (CRISPR-associated) nuclease systems have become some of the most widely applied tools for genome editing because of their simplicity, versatility, and cost-effectiveness [2]. Genome-editing technologies, including CRISPR–Cas, hold enormous potential across diverse fields, such as medical research, drug development, and agriculture. In clinical contexts, CRISPR–Cas is particularly promising for the treatment of cancers, cardiovascular diseases, sickle cell anemia, and neurodegenerative disorders [3]. In agriculture, CRISPR applications have led to notable achievements; for example, in 2020, γ-aminobutyric acid-enriched tomatoes and two genome-edited fish became the first commercially available genome-edited agricultural products [4,5]. In 2022, the US Food and Drug Administration approved the first genome-edited beef cattle, concluding that they pose no risks to humans, animals, the food supply, or the environment [6].

While several genome-edited products have already reached the market, their deployment has raised several challenges that demand urgent attention. For example, it is necessary to detect unintended insertions of foreign DNA before genome-edited products are released commercially [7,8]. In addition to the anticipated benefits of these novel technologies, it is crucial to acknowledge potential risks posed by off-target mutations—unintended nucleotide changes at nontargeted genomic sites. The off-target effects of CRISPR-Cas9 were initially observed in human cancer cell lines, where a variety of insertions, deletions, and point mutations were detected [9,10]. Notably, these unintended mutations are occasionally associated with chromosomal rearrangements, which could cause more severe toxicities than those induced by small-scale indels [11]. Recombination-activating gene nucleases, which are well-known natural nucleases, can also lead to leukemia through recombination and deletion events triggered by off-target cleavage [12,13]. Similarly, engineered nucleases with lower specificity may also contribute to genomic rearrangements. For example, transcription activator-like effector nucleases (TALENs), which were designed to edit the β-globin locus, caused large deletions and translocations in the hemoglobin gene [11]. In human cells, chromosomal translocations have been reported to follow on- and off-target double-strand breaks (DSBs) induced by CRISPR-Cas9 [14]. In agricultural species, several deletion events have been identified as possible consequences of off-target cleavage occurring in homologous sequences [15]. Although off-target mutations in crops and livestock can be mitigated through backcrossing and regeneration [16], managing off-target risks remains essential to ensure wider acceptance of genome editing as a tool for the rapid enhancement of agricultural species.

To reduce the potential risks associated with off-target mutations, methods for accurately predicting these unintended alterations in a genome are crucial. In general, both experiment-based and prediction-based approaches can be employed for this purpose. Experiment-based methods can be further divided into two categories: *in vitro* and *in vivo* techniques. *In vitro* methods include Digenome-seq [17], SITE-seq

[18], and CIRCLE-seq [19]. These methods detect off-target sites by analyzing cleaved DNA in a controlled environment. On the other hand, *in vivo* methods, such as GUIDE-seq [14], BLESS [20], and DISCOVER-seq [21], identify off-target sites within living cells. Among *in vivo* approaches, GUIDE-seq is particularly widely used. It enables the mapping of double-stranded oligodeoxynucleotide integration sites at DSBs during the end-joining repair process. The method involves unbiased amplification and next-generation sequencing of the integration sites to precisely identify DSB locations, which are considered potential off-target cleavage sites.

In parallel with experimental methods, numerous *in silico* prediction tools have been developed on the basis of various computational algorithms. One widely used program, CRISPOR, employs a score-based approach that uses a sequence aligner to identify all genomic matches [22]. The candidate regions are subsequently filtered on the basis of the presence of a protospacer adjacent motif (PAM) sequence. For each filtered match, CRISPOR calculates a cutting frequency determination (CFD) score, which incorporates position-specific mismatch tolerance weights [23]. Compared with other scoring methods, such as the MIT score, CROP-IT score, and CCTop score, the CFD score has demonstrated relatively high performance in predicting off-target sites [22]. More recently, machine learning-based algorithms have advanced off-target prediction. For example, Lin and Wong [24] employed a deep learning neural network, which achieved a higher true positive rate than did CFD and other scoring methods. Nonetheless, the performance metrics of existing prediction methods, such as precision and recall, remain suboptimal. For example, even with sophisticated deep learning models, only four out of the top 10 predicted off-target sites were confirmed as true positives [24]. Chen et al. [25] proposed a novel approach by decomposing the molecular mechanism of Cas9 into several intermolecular and intramolecular interactions, including the thermodynamic binding process between single-guide RNA (sgRNA) and DNA. However, despite these efforts, the precision–recall area under the curve (PR-AUC) values for their newly developed method, CRISOT, remained between 0.3 and 0.5 in most cases [25]. These findings suggest that factors beyond RNA–DNA interactions must be considered to achieve more accurate predictions.

The difficulty of predicting off-target sites can be attributed to several factors. For example, a study indicated that not only nearly identical sequences but also those with multiple mismatches can undergo unintended editing [26], highlighting that sequence identity alone does not fully determine off-target activity. A large-scale indel profile analysis indicated that while common patterns were generally recognized by the RNA-guided Cas9 nuclease, the precision of DNA editing varied among sites, as numerous infrequent indels were also observed [27]. Factors influencing off-target mutations induced by CRISPR–Cas systems include the number and position of mismatches between the target and potential off-target sequences, the GC-content of the target, and the delivery methods used [28]. In addition to the widely used wild-type (WT) SpCas9, a number of engineered and alternative nuclease variants with potentially different off-target activities have been developed. These include SaCas9 [29], SpCas9-HF1 [30], Cas12a [31], eSpCas9 [32], Hypa-Cas9 [33], and evoCas9 [34]. Moreover, specificity for target sequences can vary between cell lines [31]. In addition, off-target sites can be selected by chance, and such randomness may surpass a deterministic process and thereby lead to promiscuous target sequence patterns. These factors collectively contribute to the challenges in accurately predicting off-target mutations.

While genome-editing technologies offer substantial potential for targeted genetic manipulations, investigating the fundamental properties of off-target mutations remains essential. In this study, we first assessed the performance of CRISPOR's computational predictions by comparing them with off-target sites identified through the experiment-based GUIDE-seq method. Additionally, since a number of other novel methods have also been developed, CRISPOR's predictions were further validated using CRISOT, which outperforms other *in silico* methods [25]. To further explore the complexity of off-target sequence patterns, we analyzed experimental data from multiple studies. Given that various factors may influence the occurrence of off-target mutations, we examined the effects of different enzymes, cell lines, laboratory conditions, and random fluctuations. Our objective was to provide comprehensive insights into the characteristics of off-target mutations, thereby contributing to the development of safer and more effective CRISPR–Cas genome editing applications.

 

## 2. Materials and Methods

### 2.1. Preparation of GUIDE-seq data

To collect recent GUIDE-seq data, a literature search was performed on the Web of Science platform (as of April 2022). This search yielded the 500 most recent publications citing the original GUIDE-seq method development paper [14]. However, the majority of these studies neither generated novel sequence data nor made their data publicly available. As a result, in addition to the datasets from the original study [14], additional human datasets were obtained from five other studies [29,31,34–36]. Some portions of the datasets from Tan et al. [29] were excluded from the analysis because of the presence of ambiguous sgRNA sequences. S1 Table provides detailed information on the experimental conditions, enzymes, cell lines, and target sites of the 234 datasets from the six studies used in our analysis.

The FASTQ files associated with these datasets were downloaded from the International Nucleotide Sequence Database Collaboration [37]. They were trimmed using Trimmomatic (ver. 0.39) [38] with the following options: ILLU-MINACLIP:adapters.fa:2:30:10 LEADING:20 TRAILING:20 SLIDINGWINDOW:10:20 MINLEN:30. The adatpers.fa file contained all possible adapter sequences for the trimming process. Because the error rate for Illumina sequencing platforms is approximately 0.1% [39], relatively stringent threshold values were applied so that unexpectedly low-quality sequences would be excluded. FastQC (ver. 0.11.9) was used to assess the quality of the reads, both before and after trimming.

### 2.2. Data analysis for GUIDE-seq data

The analysis of the GUIDE-seq data was conducted using the programs developed by Tsai et al. [14] and bedtools (ver. 2.30.0). Our workflow followed the steps outlined in the authors' guidelines (https://github.com/tsailabSJ/guideseq). Since the GUIDE-seq datasets used in this study had already been demultiplexed prior to submission to public databases, modifications were made to the UMItag step to skip the demultiplexing process. Specifically, index files and molecular barcodes were either discarded or modified within the guideseq.py and umitag.py scripts. In the Align step, reads were aligned to the human reference genome using BWA-MEM (ver. 0.7.17-r1188) [40]. The human reference genome GRCh38 was downloaded from NCBI. The generated alignments were sorted using the sort command in Samtools (ver. 1.9) [41], and duplicate reads were removed with Picard MarkDuplicates (ver. 2.27.4). During the 'Identify' step, sequences with more than six mismatches were discarded. The remaining digested sequences, including target sequences and PAMs, were saved as a text file for further analysis. If the GUIDE-seq data were derived from technical replicates, the detected sequences were merged into a single file.

### 2.3. *In silico* prediction of off-target sites

CRISPOR was obtained from its GitHub repository (https://github.com/maximilianh/crisporWebsite) [22,42]. Appropriate guide sequences and the human reference genome were provided as inputs to the CRISPOR program. For our analysis, the following two options were specified: --skipAlign and --mm = 6. The second option, --mm = 6, allows up to six mismatches for predicting potential off-target sequences. For subsequent analysis, we selected sequences with a CFD score of 0.1 or higher, which is consistent with the threshold used in the GUIDE-seq data analysis of the original authors.

Off-target sequences identified by GUIDE-seq and predicted by CRISPOR were utilized for computing CRISOT scores [25]. Additionally, 10,000 DNA segments were randomly selected from the human reference genome, and the CRISOT scores were computed for the 18 on-target sequences used in the CRISPOR analysis. The CRISOT program (v1.0) was downloaded from the GitHub repository (https://github.com/bm2-lab/CRISOT) and executed with the 'score' option. The CRISOT scores for WT SpCas9 were subdivided into several groups, and the significance of the score differences between groups was examined by the Steel–Dwass test implemented in the pSDCFlig function of the NSM3 package in R.

## 2.4. Sequence pattern analysis

In addition to on-target sequences, several off-target sequences are typically derived from different genomic regions, although they are not necessarily identical to each other. For this study, digested sequences obtained using the same enzyme under the same conditions were aligned. The alignment patterns were quantified by calculating the relative entropy values (*R*) at each nucleotide position as follows [43]:

$$R = \sum_{n=A,T,G,C} p_n \log_2 \left( \frac{p_n}{p_g} \right)$$

where $p_n$ is the observed frequency of each nucleotide (A, T, G, or C) at a given position, and $p_g$ is the expected frequency. In this study, equal frequency was assumed for the four nucleotides, setting $p_g = 0.25$. The relative entropy calculations were performed using the rel_entr function from the SciPy (ver. 1.8.0) special function library.

The similarity of target sequence patterns between different enzymes or different experimental conditions was evaluated as follows: the relative entropy values for all nucleotide positions were sorted by magnitude, and the ranked orders of the positions were compared between enzymes or conditions. The rationale behind this approach is that, while the relative entropy is computed between observed ($p_n$) and random ($p_g$) cases, target sites should appear in a similar manner between different enzymes or between experimental conditions. In other words, if target sites are determined largely by the intrinsic nature of enzymes, their patterns are expected to deviate from the random case in a similar manner. Therefore, similar patterns should yield comparable orders of entropy values. To assess the statistical significance of the differences in these ranked orders, Spearman's rank correlation test was applied, and a 5% significance level was employed for this test. The spearmanr function from SciPy and the rstatix and corrplot packages in R were used for hypothesis testing and visualization.

The similarity between target sequence patterns was further examined using two additional metrics, the Euclidean distance and the Pearson correlation coefficient, which have been employed for motif pattern searches [44–46]. The Euclidean distance and the Pearson correlation coefficient were computed at each site, and their arithmetic means for the target sequences were calculated using a custom Python script [44–46]. The correlations among these pattern analysis methods were examined by Spearman's rank correlation test. For Euclidean distance, 5.0% of the examined pairs presented distances of ≥0.2, and this value was employed as the threshold to separate similar and divergent sequence patterns.

To generate graphical representations of sequence alignments on the basis of Shannon entropy [47,48], WebLogo3 (ver. 3.7.12) was used [49]. The aligned sequences were provided as inputs to WebLogo3, and sequence logos in PNG format were produced. The generated image files are available from https://github.com/taitoh1970/ge_off_targets.

## 3. Results

### 3.1. Off-target site identification and computational predictions

A comprehensive literature search yielded 234 GUIDE-seq datasets from six studies (S1 Table), and the FASTQ files of these datasets were trimmed using Trimmomatic (S2 Table). Since one dataset from Casini et al. [34] and 56 datasets from Chatterjee et al. [36] were technical replicates, merging them into single datasets resulted in 177 nonredundant datasets (Table 1). These datasets were generated using various enzymes, cell lines, and target sites (S1 Table), and are well suited for comparative analysis. The scale and diversity of these datasets provide a robust foundation for examining off-target sequence characteristics. Using the programs developed by Tsai et al. [14], 2,827 digested sequences, comprising both on-target and off-target sites, were identified (Table 1). This corresponds to an average of more than 16.0 digested sites per dataset. After removing on-target sites and identical off-target sequences, a total of 1,897 nonredundant off-target sites were obtained. Specifically, 927 off-target sites were detected for WT SpCas9, which served as the basis for evaluating the performance of the computational predictions.

**Table 1. Number of GUIDE-seq datasets and identified digested sequences. The numbers in the parentheses include technical replicates.**

| Study | Number of datasets | Number of digested sequences |
|---|---|---|
| Tsai et al., 2015 | 9 | 362 |
| Casini et al., 2018 | 32 (33) | 953 |
| Tan et al., 2019 | 64 | 359 |
| Choi et al., 2019 | 26 | 86 |
| Chatterjee et al., 2020 | 8 (64) | 520 |
| Zhou et al., 2022 | 38 | 547 |
| Total | 177 (234) | 2827 |

To evaluate the predictive ability of current computational tools for genome editing, we employed CRISPOR with the CFD scoring method, a widely accepted standard approach [22,42]. Owing to limitations in CRISPOR's current support for enzymes and PAM sequences, predictions were restricted to WT SpCas9. The VEGFA3 target site was excluded from this analysis because of its low specificity [50]. As a result, 4,573 nonredundant potential off-target sites with CFD scores of ≥0.1 were detected.

GUIDE-seq is an unbiased *in vivo* method with high sensitivity for identifying off-target sites, and it is widely recognized as a reliable approach. In this study, GUIDE-seq data obtained under various experimental conditions were treated as true positives, providing a benchmark for comparison with CRISPOR-predicted off-target sequences. The sequences detected by both methods were classified as true positives (TPs), those identified solely by GUIDE-seq were false negatives (FNs), and the remaining sequences reported exclusively by CRISPOR were classified as false positives (FPs). Since the inclusion of mismatches led to an explosive increase in the number of possible sequence patterns, it was expected that a significant number of false hits would be detected if a relatively large number of mismatches were allowed. In fact, although only a limited number of off-target sites were predicted when one or two mismatches were permitted, the number of predicted candidates exceeded 1,000 when five or six mismatches were allowed (Fig 1). This observation suggests that the computational predictions likely include a substantial number of erroneous sites.

To better understand the prediction performance, the predicted off-target sites were further classified on the basis of the number of mismatches (Fig 1). CRISPOR achieved relatively high recall values of 0.923, 0.825, and 0.785 for sequences with 2, 3, and 4 mismatches, respectively (Table 2). However, the recall values decreased sharply for sequences with five and six mismatches, to 0.565 and 0.025, respectively. A similar tendency was observed for false positives. While no false positives were found among the predictions with two mismatches, the precision decreased substantially for the predictions with 3–6 mismatches because of an increase in false positives (Fig 1). As a result, the $F_1$ scores remained low across these cases (Table 2).

## 3.2. Comparison between GUIDE-seq data and computational predictions

The comparison of detected sequences between CRISPOR and GUIDE-seq for WT SpCas9 revealed that CRISPOR could identify between 28% and 100% of the off-target sites detected by GUIDE-seq in different studies for various target sites (S1 Fig. and S3 Table). This analysis also revealed a significant proportion of false positives among the computational predictions, reflected in the low precision values (S3 Table). The proportion of false positives varied across different studies and target sites. For example, only 1 (1.2%) of the 84 sites predicted for HEK293site2 [14] was confirmed as a genuine off-target, whereas 86 (48.3%) of the 178 predicted sites for VEGFA2 [34] were validated by both CRISPOR and GUIDE-seq. Overall, CRISPOR appears to predict a large number of potential target sites, irrespective of the number of confirmed true positives (S1 Fig.), contributing to the observed high false-positive rates.

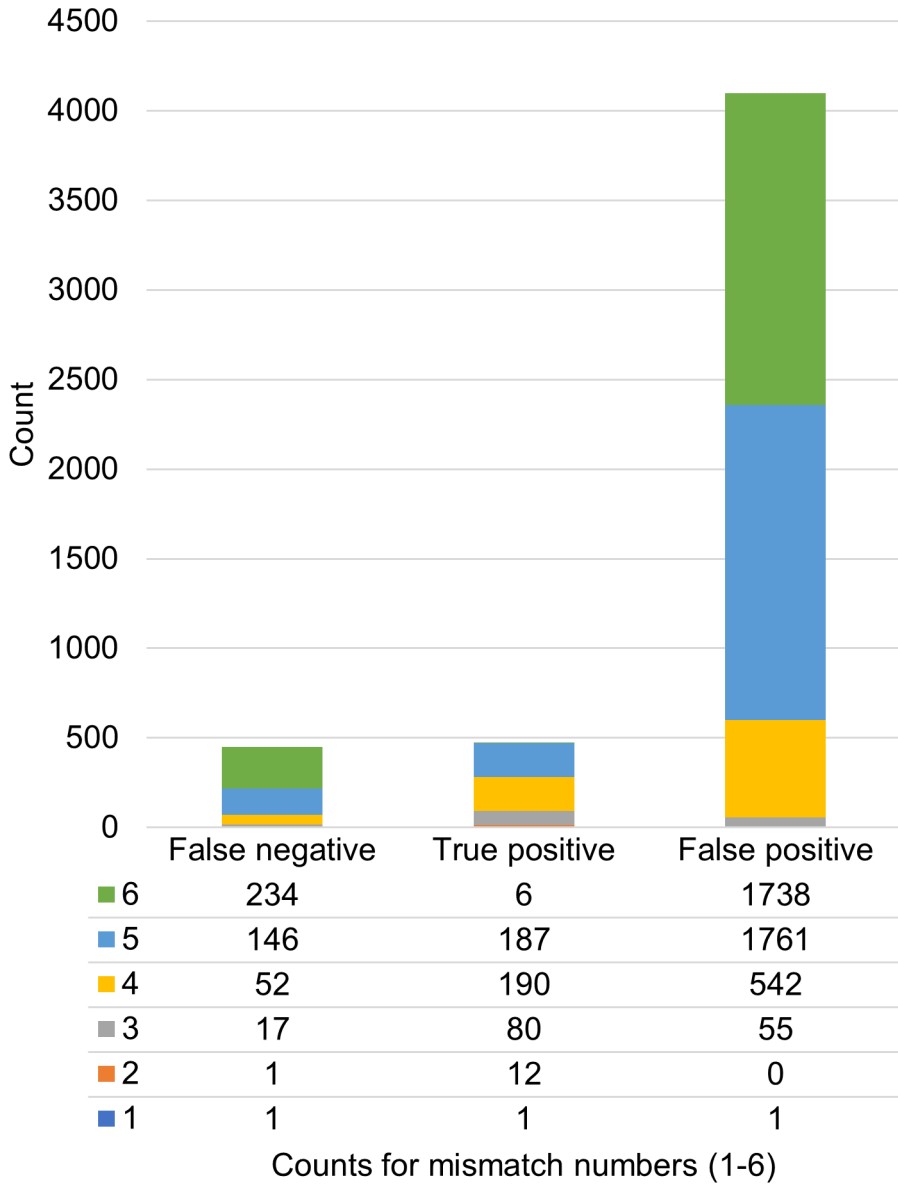

| Counts for mismatch numbers (1-6) | False negative | True positive | False positive |
|---|---|---|---|
| 6 | 234 | 6 | 1738 |
| 5 | 146 | 187 | 1761 |
| 4 | 52 | 190 | 542 |
| 3 | 17 | 80 | 55 |
| 2 | 1 | 12 | 0 |
| 1 | 1 | 1 | 1 |

**Fig 1. Number of off-target sites identified by CRISPOR and GUIDE-seq.** The sequences identified by GUIDE-seq were treated as true positives. The predicted or identified sequences were classified by the number of mismatches against their on-target sequence. Counts for each mismatch number are shown below the graph.

To further demonstrate the current computational capability of off-target predictions, another method, CRISOT, was adopted. The CRISPOR-predicted and GUIDE-seq-identified off-target sequences were divided into three groups: 4,097 FPs, 476 TPs, and 451 FNs. The CRISOT scores were subsequently computed for all of these sequences. The CRISOT score for the TP group was expected to be clearly greater than that for the FP group. In fact, the mean scores apparently differed among the groups (Fig 2), and the differences in the means were significant for all pairs of groups according to the Steel–Dwass test ($p < 0.001$). Furthermore, the CRISOT score for the TP group was clearly larger than that for the randomly selected genomic sequences (S2 Fig.), supporting the predictive performance of the method. However, the

**Table 2. Recall, precision and F$_1$ score for different mismatch numbers.** True positives, false negatives, and false positives were defined on the basis of the results of GUIDE-seq and CRISPOR.

| Mismatch number | Recall | Precision | F$_1$ Score |
|---|---|---|---|
| 1 | 0.500 | 0.500 | 0.500 |
| 2 | 0.923 | 1.000 | 0.960 |
| 3 | 0.825 | 0.593 | 0.690 |
| 4 | 0.785 | 0.260 | 0.390 |
| 5 | 0.562 | 0.096 | 0.164 |
| 6 | 0.025 | 0.003 | 0.006 |

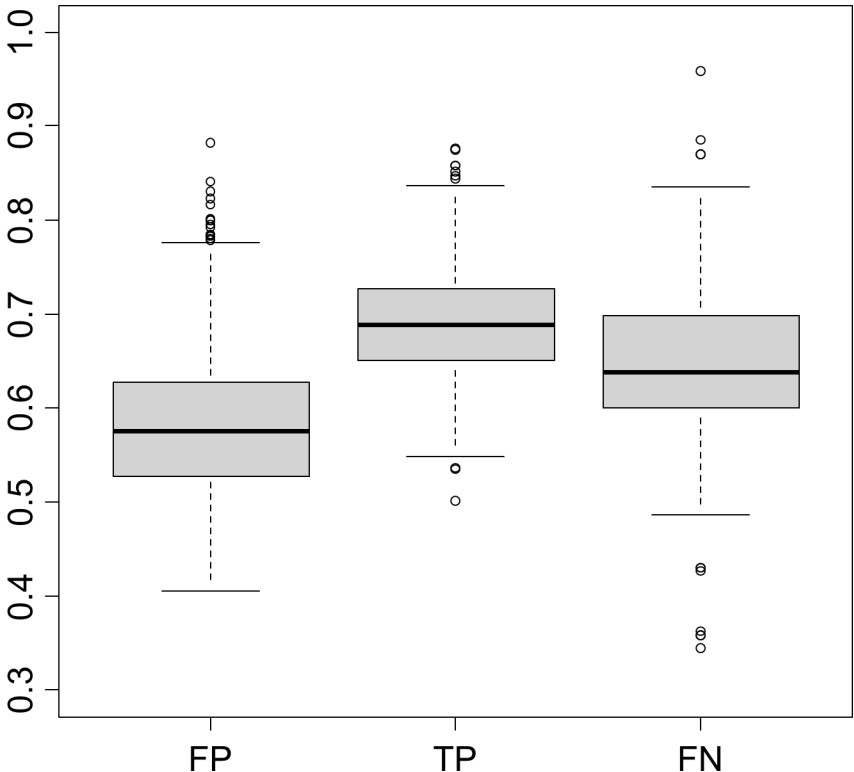

**Fig 2. Boxplots for the CRISOT scores computed for the FP, TP, and FN groups.**

distributions of the three groups clearly overlap with each other (Fig 2 and S2 Fig); therefore, true positives and true negatives are indistinguishable for a significant number of cases. Moreover, a considerable number of FPs displayed CRISOT scores of over 0.6 (S2 Fig.), indicating these off-target sequences are potentially genuine [25].

### 3.3. Detection methods for significantly similar or divergent sequence patterns around off-target sites

The challenges in accurately predicting off-target sites highlight gaps in our understanding of the characteristics underlying these sites. To investigate sequence patterns around off-target sites, we aligned the digested sequences identified by GUIDE-seq for the same target site. Sequence differences at each nucleotide position were analyzed and visualized using WebLogo3 (Figs 3 and 4). The sequence patterns for the FANCF-site6 target, investigated with Sniper-Cas9 and

**(a)**

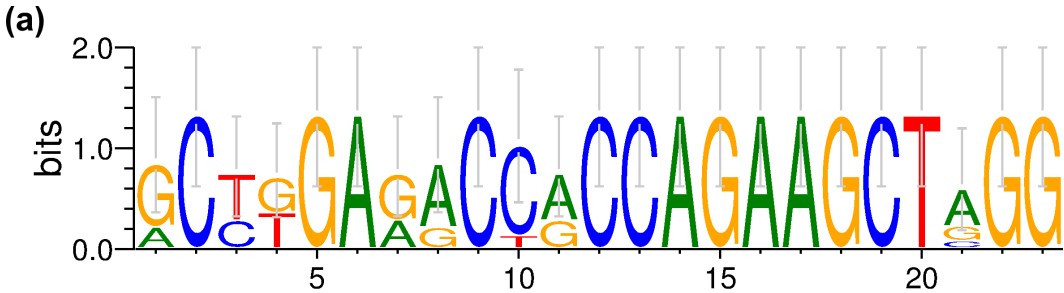

**(b)**

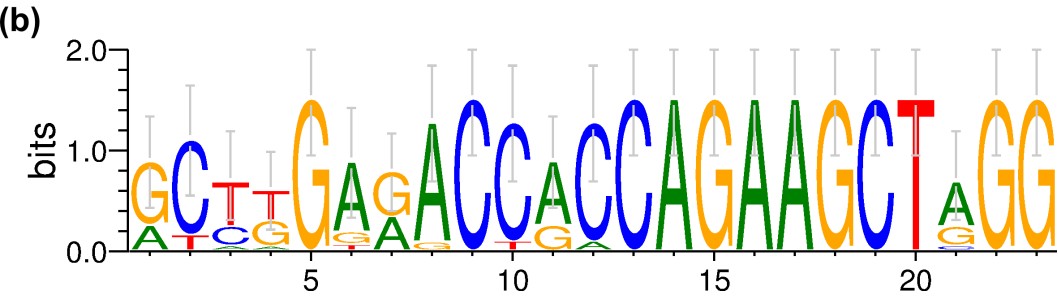

**Fig 3. WebLogo3 visualization of sequence patterns around the FANCF-site6 target site for (a) Sniper-Cas9 (8 sequences) and (b) WT SpCas9 (14 sequences).**

**(a)**

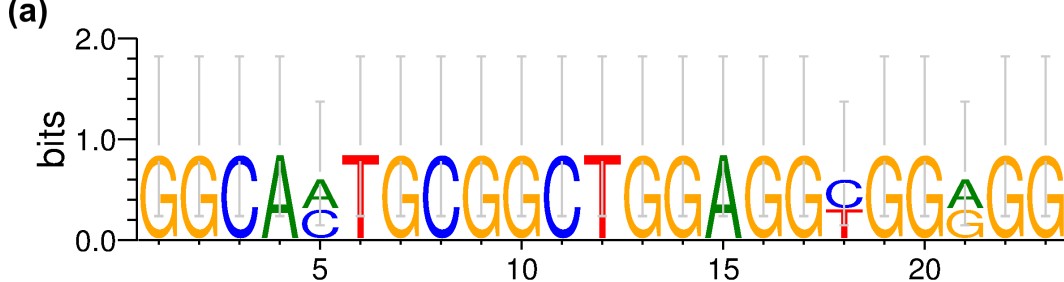

**(b)**

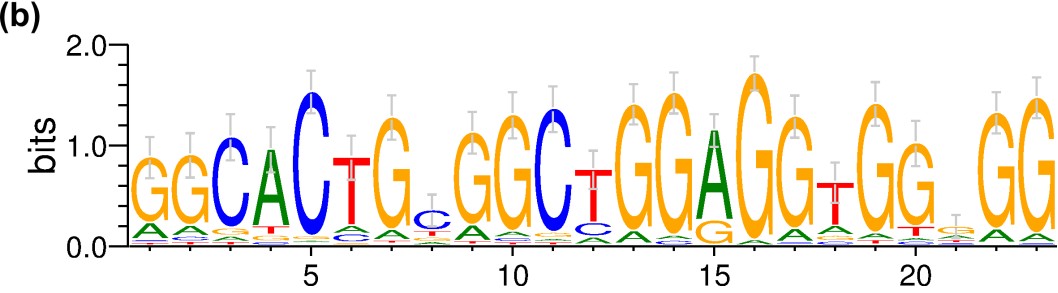

**Fig 4. WebLogo3 visualization of sequence patterns around the HEKsite4 target site for (a) SpCas9-HF1 (2 sequences) and (b) WT SpCas9 (144 sequences).**

WT SpCas9 within the same study, showed marked similarity (Fig 3). However, several other cases exhibited divergent off-target sequence patterns between experiments involving different enzyme variants. For example, the sequence patterns observed at the HEKsite4 target site differed considerably between experiments with SpCas9-HF1 and WT SpCas9 (Fig 4).

While these visual observations provided preliminary insight into similarities and differences, statistical hypothesis testing was necessary to assess their significance. The relative entropy was calculated for each nucleotide site within the alignments, and the sites were ranked by their entropy values. The assumption was that similar sequence patterns should yield similar entropy-based rankings. To test this assumption, the rank orders were compared using Spearman's rank correlation test. The analysis revealed a rank correlation coefficient of 0.71 between Sniper-Cas9 and WT SpCas9 for FANCF-site6, which was statistically significant at the 5% level. In contrast, the correlation coefficient between SpCas9-HF1 and WT SpCas9 for HEKsite4 was 0.19, indicating that there was no significant relationship ($p = 0.374$, Spearman's rank correlation test). These results align with our visual observations, confirming that the sequence patterns for FANCF-site6 were similar between the two enzymes, whereas those for HEKsite4 were divergent.

To examine the efficacy of the aforementioned method, large-scale comparisons with two additional methods based on the Euclidean distance (ED) and Pearson correlation coefficient (PCC) were performed. Spearman's rank correlation coefficients (SRCCs) for the ranked orders of entropy values and EDs were computed using 238 pairs in which the same on-target sites were digested by different enzymes or under different conditions. ED previously displayed the best performance for finding motif patterns [45], and in our comparison, it showed a clear and significant correlation with the SRCC ($\rho = -0.910$, $p < 0.001$) (Fig 5). In particular, dots are densely plotted in the lower right part (Fig 5), suggesting that similar sequence patterns are supported by both methods. Similarly, strong and significant correlations were observed between the SRCC and PCC ($\rho = 0.859$, $p < 0.001$) (S3 Fig.), and between the ED and PCC ($\rho = -0.951$, $p < 0.001$) (S4 Fig.). These results indicate that the three methods could consistently detect similar or divergent sequence patterns for the 238 pairs.

### 3.4. Analysis of differences in off-target effects among various enzymes

Off-target sequence patterns are likely influenced by several factors, including the choice of enzymes, cell lines, other experimental conditions, and random fluctuations. To explore these influences, the aforementioned 238 dataset pairs grouped by identical on-target sites were used to conduct pairwise comparisons of SRCC for the ranked relative entropy values and ED between enzymes, cell lines, or independent studies that used the same enzyme. The pairs were classified into three categories on the basis of the results of two methods, SRCC and ED: similar patterns ($p < 0.05$ for SRCC and ED $< 0.20$), semidivergent patterns ($p \geq 0.05$ for SRCC and ED $< 0.20$, or $p < 0.05$ for SRCC and ED $\geq 0.20$), and divergent patterns ($p \geq 0.05$ for SRCC and ED $\geq 0.20$). As a result, 5 cases were identified with divergent off-target sequence patterns, 18 were semidivergent, and 207 exhibited significant similarity (Table 3 and S4 Table).

As expected, several different enzymes produced divergent or semidivergent off-target mutation patterns. For example, at the CCR5 target site, the off-target sequence patterns generated by evoCas9 and SpCas9-HF1 exhibited no significant similarity ($p = 0.134$) to those produced by WT SpCas9, whereas the ED was moderately small (0.108) (S4 Table), suggesting that these enzymes created relatively unique off-target mutations (Fig 6a). However, the patterns generated by eSpCas9(1.1), evoCas9 and SpCas9-HF1 for the same target site were significantly correlated ($p < 0.05$) (Fig 6a). Similarly, at the FANCF2 target site, semidivergent off-target sequence patterns were observed between eSpCas9(1.1) and WT SpCas9 and between eSpCas9(1.1) and evoCas9, and divergent patterns were observed between WT SpCas9 and evoCas9 (Fig 6b). These findings indicate that off-target sequence patterns can vary occasionally among several enzymes. Interestingly, in many other cases, the sequence patterns of all the examined enzymes were highly similar. For example, when the VEGFA2 and VEGFA3 target sites were analyzed, the patterns generated by multiple enzymes, including WT SpCas9, eSpCas9(1.1), evoCas9, SpCas9-HF1, HIFI-Sc, HIFI-SpCas9, and Sc, were all significantly correlated (S5 Fig.).

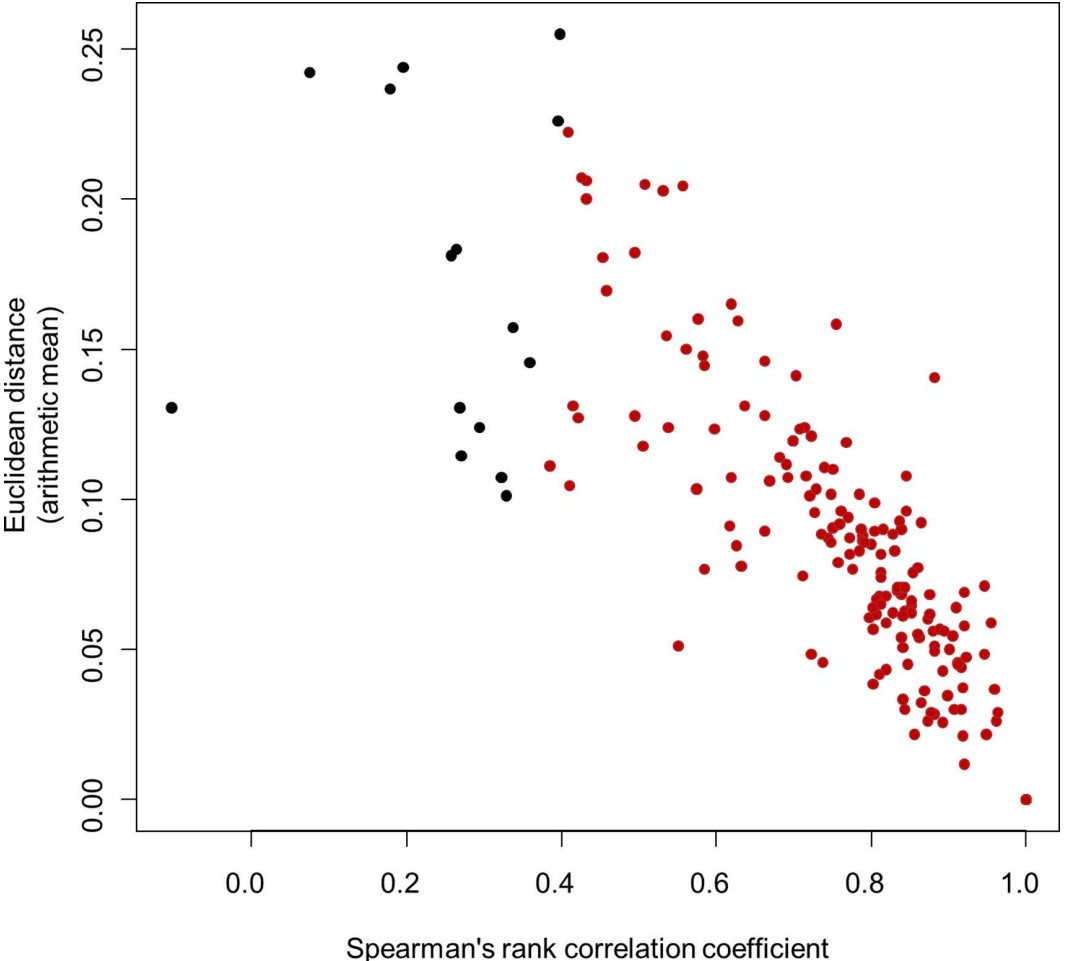

**Fig 5. Relationships between Spearman's rank correlation coefficient and the Euclidean distance for target site sequence variations among 238 pairs.** Red dots indicate significant cases ($p < 0.05$) for Spearman's rank correlation.

**Table 3. Number of cases with similar, semidivergent, and divergent sequence patterns observed among enzymes, cell lines, and studies.**

|  | Similar | Semidivergent | Divergent |
|---|---|---|---|
| Different enzymes | 200 | 18 | 5 |
| Same enzyme, different cell lines |  |  |  |
| Within the same study | 7 | 0 | 0 |
| Between different studies | 8 | 0 | 0 |

We further explored whether off-target sequence patterns were consistently similar or variable across distinct on-target sequences. For this purpose, digested sequences produced by several Cas12a variants at the POLQ2 and SITE1 target sites were compared. At the POLQ2 target site, the off-target sequence patterns for all Cas12a variants were significantly similar (Fig 7a). However, variability emerged among the five enzyme pairs when SITE1 was examined (Fig 7b). These results suggest that although multiple enzymes may yield similar off-target sequence patterns for a specific target site, the patterns can differ when the same enzymes are applied to other target sites.

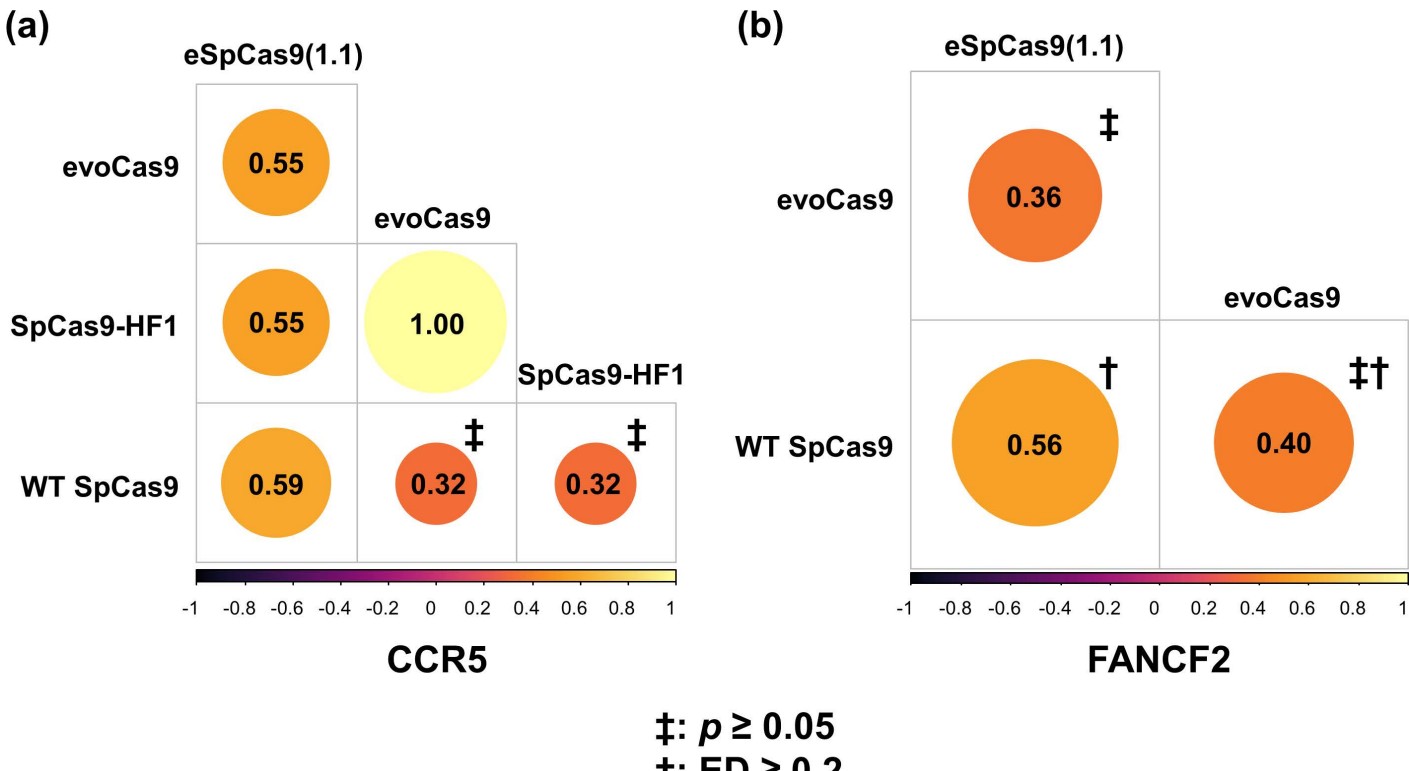

**Fig 6. Spearman's rank correlation coefficients for sequence patterns around the CCR5 and FANCF2 target sites across SpCas9 variants.** The ‡ symbol indicates an insignificant correlation ($p \geq 0.05$); the † symbol indicates a large distance (ED ≥ 0.2).

### 3.5. Differences in off-target effects under various experimental conditions

Since GUIDE-seq is an *in vivo* method, it is plausible that variation in cellular conditions could influence off-target mutation patterns. To explore this possibility, we analyzed off-target sites identified for the POLQ2 target site, which has been investigated in two different cell lines, U2OS and HEK293T [31]. Unexpectedly, the off-target sequence patterns for POLQ2 were significantly similar ($p < 0.001$), with a small ED (<0.1) across all four target sequences tested in both cell lines (S5 Table). As summarized in Table 3, no significant differences in sequence patterns were detected between the two cell lines ($p < 0.05$ for SRCC and ED < 0.2).

We further examined whether differences in laboratory environments and randomness in selecting targets might affect off-target mutation patterns. For this analysis, we compared datasets from three independent studies [14,34,36], all of which employed WT SpCas9 but used different cell lines (U2OS, HEK293T, and HEK293T/17). Contrary to our expectations, the off-target sequence patterns were highly similar across these studies ($p < 0.05$ for SRCC and ED < 0.2), despite the differences in cell lines and experimental settings (S6 Fig. and Table 2). These findings suggest that variations in laboratory conditions and cell lines and random fluctuations have minimal effects on off-target sequence patterns.

## 4. Discussion

### 4.1. Off-target sequence patterns as a consequence of the intrinsic nature of the Cas-sgRNA-DNA complex

A method based on the SRCC to compare sequence patterns of short DNA stretches was developed, and the significance of the similar and divergent sequence patterns observed in Figs 3 and 4 was proven through statistical testing. Since the

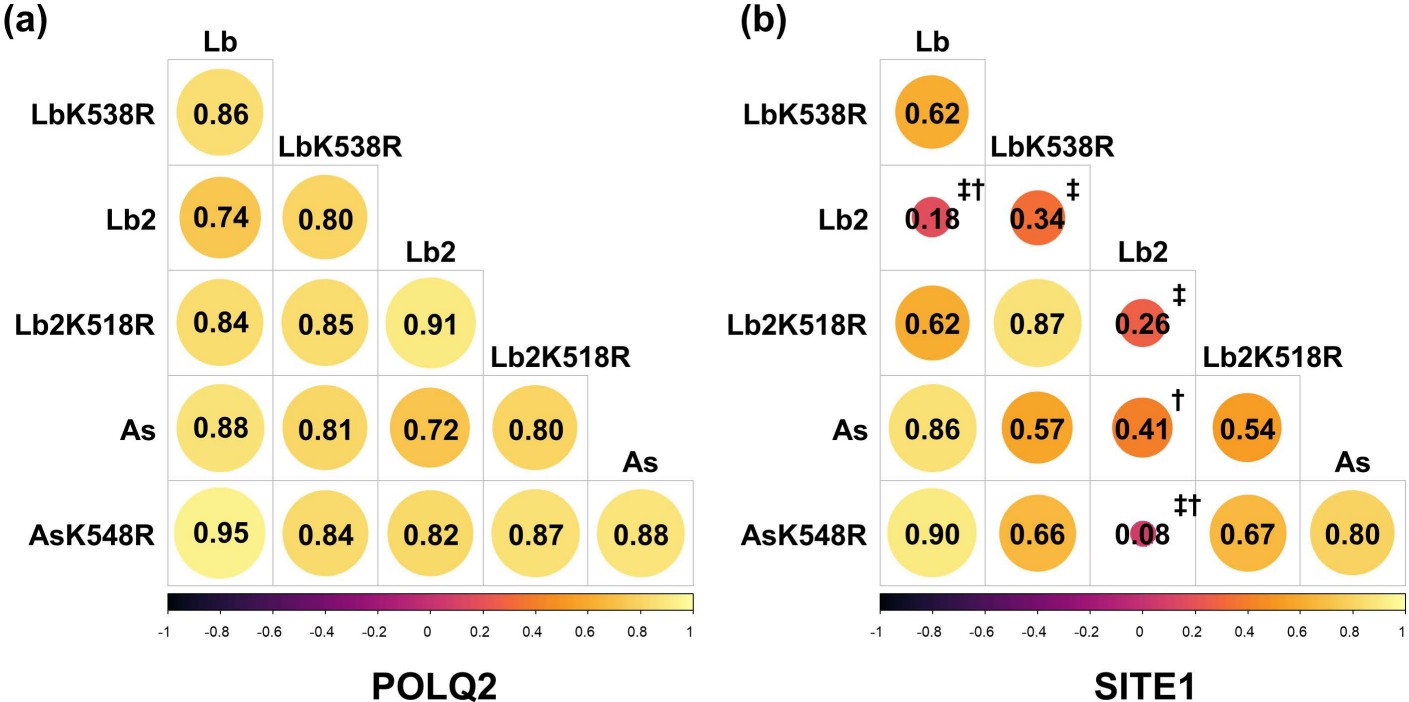

**Fig 7. Spearman's rank correlation coefficients for sequence patterns around the POLQ2 and SITE1 target sites across Cas12a variants.** The ‡ symbol indicates an insignificant correlation ($p \geq 0.05$); the † symbol indicates a large distance (ED $\geq 0.2$).

SRCC-based method assesses sequence patterns indirectly through relative entropy values, it was compared with the ED- and PCC-based methods for 238 pairs, and the results were significantly correlated (Fig 5, and S3 and S4 Figs.). As a result, 200 similar sequence patterns ($p < 0.05$ for SRCC and ED $< 0.2$) were detected (Table 3 and Fig 5). In particular, SRCC, which allows statistical significance testing, and ED were applied to analyze sequence patterns around off-target sites across three independent studies designed for the same enzyme and target site (S6 Fig.). If off-target cleavage events were purely random or influenced heavily by subtle experimental variations, different sequence patterns would be expected across these studies. However, the patterns were significantly similar, suggesting that a common underlying mechanism within the Cas-sgRNA-DNA complex plays a critical role in determining off-target sequences. The similarity observed between the two different cell lines further supports this idea (S5 Table). This consistent behavior implies that the intrinsic properties of the Cas-sgRNA-DNA complex primarily shape off-target patterns, irrespective of random fluctuations and experimental conditions. Therefore, developing prediction tools capable of accurately identifying potential off-target sites is promising when sufficient training data for a specific enzyme are available.

However, while off-target sequence patterns remain stable, the quantity of off-target mutations may vary depending on experimental conditions. For example, Fu et al. [10] and Zhou et al. [31] reported significant differences in the number of off-target sites identified between experiments using different cell lines. Despite these variations in off-target frequency, the sequence patterns remained significantly similar across conditions (S5 Table and S6 Fig.). This suggests that although the absolute numbers of off-target events may differ across cell lines, an appropriate prediction method should still be able to capture these variations accurately, providing valuable insights even when off-target site counts vary. It should also

be noted that differences in the number of off-target sequences may appear as divergent patterns (Fig 4), although such quantitative variation can be considered part of the overall pattern divergence.

### 4.2. Divergent off-target patterns derived from engineered Cas enzymes

Researchers have developed and modified CRISPR–Cas enzymes for various purposes, yet these modifications present challenges for accurate off-target prediction. Numerous studies have engineered WT SpCas9 to produce variants aimed at reducing off-target cleavage [30,32–34]. Additionally, alternative CRISPR–Cas systems derived from different bacterial species have been explored to alter PAM constraints, thereby expanding the scope of genome editing [29,31,36]. While relaxed PAM constraints are beneficial for broader applications, they also raise concerns about a potential increase in off-target mutations [51]. However, contrary to this concern, digenome-seq data analysis revealed that enzyme modification did not lead to an increase in off-target mutations [51]. This outcome suggests that novel or engineered enzymes may retain or unpredictably modify their off-target behavior. Indeed, although many enzyme pairs exhibited similar off-target sequence patterns (Fig 7a), divergent and semidivergent patterns were observed in several cases (Figs 6 and 7b and S4 Table). These findings underscore the need to thoroughly investigate off-target mutation patterns for newly engineered enzymes and suggest that computational prediction algorithms must be optimized to accommodate the unique characteristics of individual enzyme variants. Consequently, a systematic approach to evaluate off-target sequence patterns for each new enzyme is essential to ensure reliable predictions and safe application of genome-editing technologies.

### 4.3. Toward developing robust computational predictions

The molecular process of genome editing encompasses several steps, including a series of intermolecular and intramolecular interactions within the Cas–sgRNA complex [25]. While Chen et al. [25] demonstrated that precise predictions of RNA–DNA interactions could significantly improve off-target sequence inference, they also noted that many molecular mechanisms, not yet reflected in their study, should be incorporated in future models to create a more practical prediction method. Our data analysis revealed that approximately one-third of the sequences identified by GUIDE-seq were not captured by the current standard prediction tool, CRISPOR (Fig 1). Although 476 (51.3%) of the 927 off-target sites identified by GUIDE-seq for WT SpCas9 could be predicted, both recall and precision significantly decreased when the mismatch tolerance was set to five or six (Fig 1 and S3 Table). This finding aligns with a previous report on CRISPOR [22], where a false-positive rate of 0.43 was observed using a CFD score threshold of 0.023. Further analysis of CRISOT scores for CRISPOR-predicted sequences demonstrated that FNs, TPs, and FPs were indistinguishable in a significant number of the examined cases (Fig 2 and S2 Fig.).

These results indicate that, to enhance prediction performance, precise analysis is essential at each stage of genome editing, including RNA–DNA interactions specific to individual cases [25]. Notably, several recent studies have suggested that incorporating features surrounding off-target sites enhances prediction performance [52–54]. Our findings show that while some enzymes exhibit similar sequence patterns around cleavage sites, significantly different off-target sequence patterns can emerge at divergent target sites (Fig 7). This observation suggests that each independently designed sgRNA requires separate RNA–DNA interaction analyses. Additionally, the Cas-sgRNA complex itself introduces another layer of complexity, as various combinations may produce unique off-target sequence patterns. Notably, as mentioned above, newly engineered enzymes may exhibit novel off-target profiles. To optimize computational prediction, in parallel with the development of novel elaborate methods [55–57], incorporating extensive sequence pattern data generated through experimental studies into training datasets is critical.

One limitation of currently available score-based prediction methods is the lack of a consistent threshold to reliably determine if a candidate site is a true off-target site [58]. Given the impact that sgRNA and enzyme selection have on off-target sequence patterns, it is reasonable to propose that a single CFD score threshold may be inadequate. To improve prediction performance, tailored thresholds or prediction models should be developed for specific sgRNAs and Cas enzymes.

Since off-target mutations remain an unavoidable aspect of current genome-editing technologies, their occurrence must be closely monitored and minimized. Accurate *in silico* predictions tailored to specific sgRNA and Cas nuclease pairs as well as surrounding DNA features can play a crucial role in reducing these unintended mutations. In this study, we developed a method to statistically assess sequence pattern similarity and divergence. The findings obtained using this method provide a foundation for developing a framework aimed at enhancing computational methods for accurate off-target sequence prediction tailored to individual enzymes and target sites. Although we conducted large-scale data analysis to discover the characteristics of off-target sequences, an increasing amount of experimental data is now becoming available through GUIDE-seq and other effective methods. It is expected that further analysis of such data will lead to the identification of precise biological mechanisms of off-target mutation generation, and if these mechanisms are validated experimentally, this information can be used to improve computational prediction models in the future.

## Supporting information

**S1 File.    S1 Fig. Bar plots showing the number of off-target sites predicted exclusively by CRISPOR (FP, false positives), identified solely by GUIDE-seq (FN, false negatives), and shared between both methods (TP, true positives). S2 Fig. Distributions of CRISOT scores for the FP, TP, and FN groups.** The mean and standard deviation for each group are shown. **S3 Fig. Relationship between Spearman's rank correlation coefficient and the Euclidean distance.** Red dots indicate cases with statistically significant Spearman's rank correlations. **S4 Fig. Relationship between the Euclidean distance and the Pearson correlation coefficient.** Red dots indicate cases with statistically significant Spearman's rank correlations. **S5 Fig. Spearman's rank correlation coefficients for sequence patterns surrounding the VEGFA2 and VEGFA3 target sites across various Cas enzymes.** All correlations were statistically significant with small EDs (<0.2). **S6 Fig. Spearman's rank correlation coefficients for sequence patterns surrounding the EMX1 and VEGFA3 target sites using WT SpCas9 across three different studies and cell lines.** All correlations were statistically significant with small EDs (<0.2).
(ZIP)

**S2 File.    S1 Table. Details of the GUIDE-seq datasets used in this study. S2 Table. Accession numbers of the sequence data used in this study. S3 Table. Recall and precision values for different target sites. S4 Table. Divergent and semidivergent target sequence patterns. S5 Table. Similarity of target patterns for POLQ2 between the HEK293T and U2OS cell lines.**
(ZIP)

## Acknowledgments

We thank Chwan-Yang Hong and Yu-Chang Tsai for their valuable comments on this study in its early stages and Yoshiharu Y. Yamamoto for his comments on sequence pattern analysis.

## Author contributions

**Conceptualization:** Takeshi Itoh.

**Formal analysis:** Celine Kurniawan.

**Funding acquisition:** Takeshi Itoh.

**Investigation:** Celine Kurniawan.

**Methodology:** Takeshi Itoh.

**Project administration:** Takeshi Itoh.

**Supervision:** Takeshi Itoh.

**Validation:** Takeshi Itoh, Celine Kurniawan.

**Visualization:** Celine Kurniawan.

**Writing – original draft:** Celine Kurniawan.

**Writing – review & editing:** Takeshi Itoh.

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
