## [Decision Letter · Decision Letter 0]

Dear Dr. Itoh,

**Your manuscript was reviewed by four experts. While they found it potentially interesting, they raised some concerns. Please revise it according to their suggestions.**

We look forward to receiving your revised manuscript.

Kind regards,

Hodaka Fujii, M.D., Ph.D.

Academic Editor

PLOS ONE

**Journal Requirements:**

This work was supported by the Office of Research and Development New Faculty Grant (111L7472), National Taiwan University, and Grant 111-2313-B-002-006-MY3 from the National Science and Technology Council, Republic of China (Taiwan).

Reviewers' comments:

Reviewer's Responses to Questions

**Comments to the Author**

1. Is the manuscript technically sound, and do the data support the conclusions?

Reviewer #1: Yes

Reviewer #2: No

Reviewer #3: Partly

Reviewer #4: No

2. Has the statistical analysis been performed appropriately and rigorously?

Reviewer #1: Yes

Reviewer #2: No

Reviewer #3: No

Reviewer #4: No

3. Have the authors made all data underlying the findings in their manuscript fully available?

Reviewer #1: Yes

Reviewer #2: Yes

Reviewer #3: Yes

Reviewer #4: Yes

4. Is the manuscript presented in an intelligible fashion and written in standard English?

Reviewer #1: Yes

Reviewer #2: Yes

Reviewer #3: No

Reviewer #4: No

**Reviewer #1: ** The manuscript PONE-D-24-52272 analyses GUIDE-Seq data from 234 datasets from six previous different studies, with different cell lines, enzymes and target sites to compare off-target sequence patterns in genome editing. Moreover, it evaluates the predictions of off-target sites by CRISPOR against those experimental data.

The draft is well written and the methods used seem correct.

There are a few minor problems to address before publication.

Introduction

L.72-73: "in vitro and in vitro techniques" please, correct.

Results

Paragraphs 3.3 and 3.4 could be merged.

L.303-304 "In total, 16 instances were identified with distinct off-target sequence

patterns, while 207 cases exhibited significant similarity (Table 2)." Please clarify "instances". Do you mean triples of (target site, enzime1, enzime2)? Similarly, clarify Table 2.

Fig.2 and Fig.3: Please add the number of sequences aligned in each case.

Table 2: Please clarify the caption.

**Reviewer #2:**  The authors of the manuscript under review with the title "Off-target sequence variations

driven by the intrinsic properties of the Cas-sgRNA-DNA complex in genome editing"

evaluate the performance for off-target site predictions with the program CRISPOR on six

independent GUIDE-seq datasets in genome editing applications. They furthermore

compare off-target site sequences and find specific sequence motifs, which seem to be

dependent on the used enzyme, cell-lines and other factors. Off-target site cleavages must

be prohibited for many applications to limit genome editing only to the intended sites for safe

use in the future.

The general research question is of great relevance and the principle approach to assess

the benchmark between in silico predictions with in vivo or in vitro derived high throughput

sequencing data is convincing.

However, I do have three major concerns with the current status of the manuscript:

a) The authors report about their extensive literature research, which unfortunately

resulted in only 6 studies that made their sequencing data publicly available. It would

be great to base the assessment on a more diverse data foundation, but I don't see

any means how the authors should achieve that other than repeating the literature

research, which was done back in 2022. Chances are high that further GUIDE-seq

data have been submitted in between.

On the other hand, the authors limit themselves to only ONE computational

prediction tool, namely CRISPOR, although they state that "numerous in silico

prediction tools have been developed" (line 82). Alternatives to CRISPOR are not

even discussed in the manuscript. I expect the authors to re-do their analysis also for

those other software - which should be well within their reach.

b) The authors did align off-target site sequences and quantified the alignment pattern

by relative entropy. They formulate the background assumption that "equal frequency

was assumed for the four nucleotides, setting p_g = 0.25" (line 182).

I disagree with this assumption! If I am not mistaken, you design one target

nucleotide sequence, through the guide RNA, in CRISPR-Cas. Thus, I'd "expect" all

targeted sites to be this guide RNA sequence. This should also apply to off-target

sites. Therefore, I'd assume p_g to be sequence position specific, namely 1.0 for the

nucleotide of the guide and 0 otherwise - maybe add some uncertainty to avoid 0

probabilities, i.e. 0.97, 0.01, 0.01 and 0.01 for the four bases.

c) I have even greater concerns with the authors' similarity metric to compare multiple

off-target site "patterns". It is purely based on relative entropy, which is independent

of sequence content. This could lead to extremely different sequence patterns that

are considered identical. For example, take the pattern of Figure 2, panel (a). Entropy

of positions 12 to 20 is basically at maximum, i.e. only these bases have been

observed. If we replace the bases CCAGAAGCT with TTTTTTTTA, which is

obviously very different, the entropy wouldn't change at all!

I am therefore convinced that all downstream conclusions are flawed!

Minor issues:

d) The presentation of the number of studies / datasets is inconsistent and hard to

understand. Authors speak about one "plus five additional human datasets" (line 132)

in the Materials and Methods section, "three independent studies" (line 349) in the

Discussion but present 234 "datasets" and 2827 "digested sequences" in Table 1.

e) The pattern construction and comparison strongly reminds me of transcription factor

binding site prediction and the rich research body on this topic. Authors might read

up methodology in this multiple decades ago established field to carry over best

practices to their application.

f) The table in Figure 1 and the stacked bar-chart is duplicating information.

Supplementary Table 3, on the other hand, holds valuable information which I would

not hide in the supplement but directly present in Figure 1. Maybe even merge with

supplementary Figure 1 or create a multi panel figure.

g) I think statements like "These results align with our visual observations, confirming

that the sequence patterns for FANCF-site6 were similar between the two enzymes,

whereas those for HEKsite4 were dissimilar." (line 279) are probably invalidated with

a proper similarity metric and I don't like the fact that authors only evaluated their

metric "visually" on just two examples. A more thorough investigation should have

brought up the sequence independence problem - see my issue c).

h) I do not directly see how one could test the significance of the similarity, e.g. line 291,

without a proper background model of pattern similarities, a density function or a

permutation approach. Depending on the clustering of off-site target sequences and

their multiple sequence alignment, it is likely that we have a high false negative rate

and thus a very incomplete ranked list of patterns.

In conclusion, my recommendation is to only accept this manuscript if authors provide major

revisions. The methodology is flawed, but the general research question is worth being

pursued.

**Reviewer #3:**  The authors omit several studies about the target-specific precision of CRISPR-Cas. For example, how does the study presented in the manuscript relate to the findings in the article: 10.1016/j.molcel.2018.11.031?

In the Methods section, the authors include many unnecessary details, such as instructions on which packages need to be installed to run the program CRISPOR. However, they fail to provide essential details such as the library size of the datasets and the outcomes of data cleaning. Additionally, the criteria for trimming the first and last 20 nucleotides of reads seem overly strict and warrant justification.

In the Results section, Figure 1 includes information about detected off-target sites. However, the figure is not well-explained, and the results are insufficiently commented on in the text.

Table 2 presents the frequency of sequence patterns in different cell lines, enzymes, and studies. The table could provide a more detailed exploration of the differences between enzymes, particularly highlighting whether the same or different cell lines and studies were involved. For the same enzyme, dividing data into different studies seems unnecessary, as the patterns appear similar in both cases.

What criteria are used to determine whether a correlation is significant? This is not explained in the text, particularly concerning Figures 4 and 5.

In the Introduction section, the phrase "in vitro" appears twice, but one instance should be replaced with "in vivo" to correctly describe both methodologies.

The citation style is inconsistent throughout the manuscript. Some articles are cited by numbers, while others are referenced by name and year, which creates confusion. A consistent citation format should be adopted.

The manuscript also requires a thorough review for grammatical errors, typos, and improvements in English usage to enhance clarity and readability.

**Reviewer #4:**  The authors have presented a study on evaluating the off-target sequence patterns of the Cas-sgRNA-DNA complex in determining cleavage sites by collecting various experimental off-target data and conducting a large-scale computational analysis, aiming for providing insights for a better understanding of off-target mutations and developing more accurate computational off-target prediction methods. The topic is important. However, the experiments look insufficient for such a bioinformatics study. The manuscript could be revised for further consideration. Here are some of the comments:

1. The authors provided detailed analysis about sequence patterns. But they didn’t dig up the biologically significance of these patterns. I suggest the authors deeply investigating these sequence patterns to see how they affect the cleavage process of CRISPR/Cas system. Specifically, the results of WebLogo visualization of sequence patterns among different enzymes show various patterns in off-target mutations, but the authors did not provide detailed analysis or give the universal rule of these observations. Some examples can be discussed as case studies for biological insights in the discussion section.

2. The authors mentioned several deep learning-based off-target prediction methods in the Introduction section. I suggest them follow the latest research reports, such as:

[1] Yaish O, Orenstein Y. Generating, modeling and evaluating a large-scale set of CRISPR/Cas9 off-target sites with bulges, Nucleic Acids Research 2024;52(12):6777-90.

[2] Vora DS, Bhandari SM, Sundar D. DNA shape features improve prediction of CRISPR/Cas9 activity, Methods 2024;226:120-26.

[3] Toufikuzzaman M, Hassan Samee MA, Sohel Rahman M. CRISPR-DIPOFF: an interpretable deep learning approach for CRISPR Cas-9 off-target prediction, Brief Bioinform 2024;25(2).

[4] Sinha S, Sun J, Guo J et al. CRISPR-M: Predicting sgRNA off-target effect using a multi-view deep learning network, PLOS Computational Biology 2024;20(3).

[5] Özden F, Minary P. Learning to quantify uncertainty in off-target activity for CRISPR guide RNAs, BioRxiv 2024:2023.06.02.543468.

[6] Luo Y, Chen Y, Xie H et al. Interpretable CRISPR/Cas9 off-target activities with mismatches and indels prediction using BERT, Comput Biol Med 2024;169.

3. The English should be sanitized.

**Do you want your identity to be public for this peer review?** For information about this choice, including consent withdrawal, please see our Privacy Policy

Reviewer #1: **Yes: ** Serena Rosignoli

Reviewer #2: No

Reviewer #3: No

Reviewer #4: No

---

## [Author Response · Author response to Decision Letter 1]

1 May 2025

Response to Reviewers

We sincerely appreciate the four reviewers for their thoughtful and thorough evaluation of our manuscript. Their insightful comments were highly valuable and have significantly contributed to improving the quality of our work. We have revised the manuscript accordingly and provide a point-by-point response below.

During the course of an extensive reanalysis of the datasets, we discovered that on-target sequences had been inadvertently included in the GUIDE-seq data only, which affected S1 Fig. and S3 Table. However, we would like to note that since only one true positive was removed from each GUIDE-seq data, the overall results remained largely unchanged and did not alter our main conclusions.

> Reviewer #1: The manuscript PONE-D-24-52272 analyses GUIDE-Seq data from 234 datasets from six previous different studies, with different cell lines, enzymes and target sites to compare off-target sequence patterns in genome editing. Moreover, it evaluates the predictions of off-target sites by CRISPOR against those experimental data.

The draft is well written and the methods used seem correct.

There are a few minor problems to address before publication.

We greatly appreciate the reviewer’s positive evaluation of our study.

> Introduction

L.72-73: "in vitro and in vitro techniques" please, correct.

This typo has been corrected.

> Results

Paragraphs 3.3 and 3.4 could be merged.

We intended for paragraph 3.3 to describe the developed method, while paragraph 3.4 presents in-depth data analyses. We believe that keeping these sections separate helps readers better understand the structure of our manuscript. To clarify this distinction, we have revised the section titles as follows:

3.3. Detection methods for significantly similar or divergent sequence patterns around off-target sites

3.4. Analysis of differences in off-target effects among various enzymes

> L.303-304 "In total, 16 instances were identified with distinct off-target sequence patterns, while 207 cases exhibited significant similarity (Table 2)." Please clarify "instances". Do you mean triples of (target site, enzime1, enzime2)? Similarly, clarify Table 2.

The reviewers understanding is essentially correct. Here, "instance" refers to a comparison of datasets between two distinct experiments with a specific target site. To improve readability, we have rewritten the indicated sentence.

> Fig.2 and Fig.3: Please add the number of sequences aligned in each case.

The numbers have been added in the figure legends.

> Table 2: Please clarify the caption.

To improve clarity, we have revised the caption for Table 2. Please note that Table 2 has been renamed as Table 3 in the revised manuscript.

> Reviewer #2: The authors of the manuscript under review with the title "Off-target sequence variations driven by the intrinsic properties of the Cas-sgRNA-DNA complex in genome editing" evaluate the performance for off-target site predictions with the program CRISPOR on six independent GUIDE-seq datasets in genome editing applications. They furthermore compare off-target site sequences and find specific sequence motifs, which seem to be dependent on the used enzyme, cell-lines and other factors. Off-target site cleavages must be prohibited for many applications to limit genome editing only to the intended sites for safe use in the future.

The general research question is of great relevance and the principle approach to assess the benchmark between in silico predictions with in vivo or in vitro derived high throughput sequencing data is convincing.

We sincerely appreciate the reviewer's recognition of the importance of our study.

> However, I do have three major concerns with the current status of the manuscript:

a) The authors report about their extensive literature research, which unfortunately resulted in only 6 studies that made their sequencing data publicly available. It would be great to base the assessment on a more diverse data foundation, but I don't see any means how the authors should achieve that other than repeating the literature research, which was done back in 2022. Chances are high that further GUIDE-seq data have been submitted in between.

For this study, we manually examined hundreds of papers that cited the original GUIDE-seq paper. Unfortunately, due to time-constraints, it is not feasible for us to repeat this process within a realistic timeframe. Nevertheless, our study includes 177 non-redundant datasets derived from six independent studies (Table 1, S1 and S2 Tables), which encompass a range of enzymes, cell lines, and target sites. We believe that they are sufficiently large and diverse to support the conclusions drawn in the manuscript. That said, we fully understand the reviewer's concern. To further reassure readers of the robustness of our datasets, we have added S2 Table, which clearly lists the datasets analyzed. Additionally, we have expanded the last paragraph of the discussion to address this point more explicitly.

> On the other hand, the authors limit themselves to only ONE computational prediction tool, namely CRISPOR, although they state that "numerous in silico prediction tools have been developed" (line 82). Alternatives to CRISPOR are not even discussed in the manuscript. I expect the authors to re-do their analysis also for those other software - which should be well within their reach.

In response to the reviewer’s suggestion, we have now included an additional computational prediction tool, CRISOT, alongside CRISPOR. Notably, CRISOT has been reported to outperform other major methods (Chen et al., 2023). However, in our analysis, it was still difficult to clearly distinguish between true and false hits (Fig. 2 and S2 Fig.). We would like to note that, in this examination of software tools, our primary aim was to illustrate the current state of available standard programs, rather than to conduct an exhaustive comparison across multiple programs. Therefore, we believe that the inclusion of CRISOT sufficiently addresses the reviewer’s concern within the scope of this work.

> b) The authors did align off-target site sequences and quantified the alignment pattern by relative entropy. They formulate the background assumption that "equal frequency was assumed for the four nucleotides, setting p_g = 0.25" (line 182). I disagree with this assumption! If I am not mistaken, you design one target nucleotide sequence, through the guide RNA, in CRISPR-Cas. Thus, I'd "expect" all targeted sites to be this guide RNA sequence. This should also apply to off-target sites. Therefore, I'd assume p_g to be sequence position specific, namely 1.0 for the nucleotide of the guide and 0 otherwise - maybe add some uncertainty to avoid 0 probabilities, i.e. 0.97, 0.01, 0.01 and 0.01 for the four bases.

In this part of our analysis, we aimed to quantify the "distance" between the observed patterns and those expected under a random scenario. If the Cas-sgRNA-DNA complexes exhibit similar properties, we would expect that their distances from the random case would be similar. However, if two observed patterns deviate in different directions from the random baseline, such differences can be detected using an appropriate statistical method. To capture this, we employed Spearman's rank correlation test. To prevent potential misunderstanding, we have revised the Methods section for clarity. Additionally, as we describe later in this reply, our method showed a significant correlation with two independent metrics, which supports the validity of our approach.

> c) I have even greater concerns with the authors' similarity metric to compare multiple off-target site "patterns". It is purely based on relative entropy, which is independent of sequence content. This could lead to extremely different sequence patterns that are considered identical. For example, take the pattern of Figure 2, panel (a). Entropy of positions 12 to 20 is basically at maximum, i.e. only these bases have been observed. If we replace the bases CCAGAAGCT with TTTTTTTTA, which is obviously very different, the entropy wouldn't change at all!

I am therefore convinced that all downstream conclusions are flawed!

We fully acknowledge the importance of thoroughly evaluating the effectiveness of our similarity metric. In response to the reviewer’s concern—as well as the related point raised in comment e)—we incorporated two metrics for comparison: Euclidean distance, which demonstrated the best performance for motif detection in Gupta et al. (2007), and the Pearson correlation coefficient, which is the default method used in the Tomtom web service (https://meme-suite.org/meme/tools/tomtom). Our analysis of 238 dataset pairs revealed strong correlations among the methods, indicating that our proposed approach is consistent with both Euclidean distance and Pearson correlation. Based on this, we are confident that our method is appropriate for investigating sequence pattern similarity.

In the revised manuscript, we report results using both our method and Euclidean distance, given its demonstrated effectiveness in prior research (Gupta et al., 2007).

Although we believe the extreme cases mentioned by the reviewer may be uncommon, we have addressed such scenarios by introducing a "semidivergent" category in Table 3 and S4 Table. This category includes 7 cases that were not significant by Spearman’s correlation but showed relatively large Euclidean distances.

> Minor issues:

> d) The presentation of the number of studies / datasets is inconsistent and hard to understand. Authors speak about one "plus five additional human datasets" (line 132) in the Materials and Methods section, "three independent studies" (line 349) in the Discussion but present 234 "datasets" and 2827 "digested sequences" in Table 1.

We appreciate the reviewer’s comment and have thoroughly revised the main text to clarify the presentation of the studies and datasets. In addition, we now provide S1 and S2 Tables to ensure there is no ambiguity regarding the number and sources of datasets used in our analysis.

> e) The pattern construction and comparison strongly reminds me of transcription factor binding site prediction and the rich research body on this topic. Authors might read up methodology in this multiple decades ago established field to carry over best practices to their application.

We appreciate the reviewer’s insightful observation. In response, we consulted relevant literature and adopted the use of the Euclidean distance and the Pearson correlation coefficient for our pattern comparisons (Gupta et al., 2007; Choi et al., 2004; Pietrokovski, 1996). For additional details and results, please refer to our response to comment c).

> f) The table in Figure 1 and the stacked bar-chart is duplicating information. Supplementary Table 3, on the other hand, holds valuable information which I would not hide in the supplement but directly present in Figure 1. Maybe even merge with supplementary Figure 1 or create a multi panel figure.

We appreciate the reviewer’s suggestion. In our view, the stacked bar chart and the table in Fig. 1 serve complementary purposes and together provide readers with a clearer understanding of the data. In response to the comment, we have moved the former S3 Table into the main text as new Table 2. However, it is difficult to integrate Fig. 1 and Table 2 into a single figure in a clear and readable format. Therefore, we have opted to present them separately while keeping both in the main manuscript.

> g) I think statements like "These results align with our visual observations, confirming that the sequence patterns for FANCF-site6 were similar between the two enzymes, whereas those for HEKsite4 were dissimilar." (line 279) are probably invalidated with a proper similarity metric and I don't like the fact that authors only evaluated their metric "visually" on just two examples. A more thorough investigation should have brought up the sequence independence problem - see my issue c).

As noted in our response to comment c), we have now conducted a large-scale comparison of our metric with two widely used alternatives, following the reviewer’s suggestion. We are pleased to know that this broader evaluation supports the validity of our approach beyond the initial visual inspection of two examples.

> h) I do not directly see how one could test the significance of the similarity, e.g. line 291, without a proper background model of pattern similarities, a density function or a permutation approach. Depending on the clustering of off-site target sequences and their multiple sequence alignment, it is likely that we have a high false negative rate and thus a very incomplete ranked list of patterns.

> In conclusion, my recommendation is to only accept this manuscript if authors provide major revisions. The methodology is flawed, but the general research question is worth being pursued.

In our study, the target sequences and their alignments could be determined with high confidence and little ambiguity. Consistent patterns of similarity were observed across three independent metrics, suggesting that the likelihood of false negatives is low.

As the title of the manuscript indicates, our focus is on highlighting that target sequences are largely determined by the intrinsic properties of the Cas-sgRNA-DNA complex, leading to conserved sequence features. We do not intend to emphasize dissimilarity, although we acknowledge that divergent cases may be of interest and warrant further discussion. To avoid confusion, we have chosen to replace the term dissimilarity with more appropriate alternatives such as divergent and semidivergent throughout the manuscript.

> Reviewer #3: The authors omit several studies about the target-specific precision of CRISPR-Cas. For example, how does the study presented in the manuscript relate to the findings in the article: 10.1016/j.molcel.2018.11.031?

We appreciate the reviewer for bringing this important reference to our attention. We have now cited the indicated study in the main text.

> In the Methods section, the authors include many unnecessary details, such as instructions on which packages need to be installed to run the program CRISPOR.

In response to the reviewer’s suggestion, we have removed several sentences from the Methods section to eliminate unnecessary technical details and improve readability.

> However, they fail to provide essential details such as the library size of the datasets and the outcomes of data cleaning.

We agree that this information is important. To address this issue, we created a new table (S2 Table) that provides detailed descriptions of all sequence datasets. Together with S1 Table, we believe these additions offer sufficient transparency regarding the materials used in this study.

> Additionally, the criteria for trimming the first and last 20 nucleotides of reads seem overly strict and warrant justification.

We appreciate the reviewer’s comment. The datasets used in this study were generated by various research groups, and we observed that some contained a substantial number of low-quality sequences at the read ends. To ensure data reliability, we applied trimming based on a quality threshold of Q20, which corresponds to a <1% error rate. While this may seem relatively stringent, it remains reasonable given that the average error rate for Illumina sequencers is approximately 0.1% (Schirmer et al., 2016). We have now clarified this rationale in the Methods section.

> In the Results section, Figure 1 includes information about detected off-target sites. However, the figure is not well-explained, and the results are insufficiently commented on in the text.

We thank the reviewer for this helpful suggestion. In response, we have reorganized the relevant parts of the main text and added more detailed explanations in both the figure legend and the Results section to improve clarity and ensure that the figure is fully understood.

> Table 2 presents the frequency of sequence patterns in different cell lines, enzymes, and studies. The table could provide a more detailed exploration of the differences between enzymes, particularly highlighting whether the same or

---

## [Decision Letter · Decision Letter 1]

Dear Dr. Itoh,

Thank you for submitting your manuscript to PLOS ONE. After careful consideration, we feel that it has merit but does not fully meet PLOS ONE’s publication criteria as it currently stands. Therefore, we invite you to submit a revised version of the manuscript that addresses the points raised during the review process.

We look forward to receiving your revised manuscript.

Kind regards,

Hodaka Fujii, M.D., Ph.D.

Academic Editor

PLOS ONE

Journal Requirements:

Reviewers' comments:

Reviewer's Responses to Questions

**Comments to the Author**

Reviewer #1: (No Response)

Reviewer #2: (No Response)

Reviewer #3: All comments have been addressed

2. Is the manuscript technically sound, and do the data support the conclusions?

Reviewer #1: Partly

Reviewer #2: Partly

Reviewer #3: Yes

3. Has the statistical analysis been performed appropriately and rigorously?

Reviewer #1: Yes

Reviewer #2: N/A

Reviewer #3: Yes

4. Have the authors made all data underlying the findings in their manuscript fully available?

Reviewer #1: Yes

Reviewer #2: (No Response)

Reviewer #3: Yes

5. Is the manuscript presented in an intelligible fashion and written in standard English?

Reviewer #1: Yes

Reviewer #2: Yes

Reviewer #3: Yes

Reviewer #1: In this revised version of the manuscript, the authors addressed the comments of the reviewers, however some points require further considerations.

Major points:

23 "We developed a method to assess sequence pattern similarity and diversity between off-target sites. This method is based on a comparison of ordered relative entropy values for aligned target sequences, and it was compared with two other methods on the basis of Euclidean distance and the Pearson correlation coefficient. The three methods demonstrated clear correlations, indicating their validity."

366 "These results indicate that the three methods could consistently detect similar or divergent sequence patterns for the 238 pairs" It is unclear what is the advantage of the new method based on ordered relative entropy.

130 "In this study, we first assessed the performance of CRISPOR's computational predictions by comparing them with off-target sites identified through the experiment-based GUIDE-seq method. Additionally, since a number of other novel methods have also been developed, CRISPOR’s predictions were further validated using CRISOT, which outperforms other in silico methods [25]"

192 "Off-target sequences identified by GUIDE-seq and predicted by CRISPOR were utilized for computing CRISOT scores [25]"

If CRISOT outperforms other in silico methods, including CRISPOR, the prediction analyses should have been done using CRISOT; and also the 'Comparison between GUIDE-seq data and computational predictions" of 3.2 and Fig.1. Instead, CRISOT was used only to score the off-targets in common between CRISPOR and GUIDE-Seq. It is also possible that CRISOT could have found off targets undetected by both the other two methods.

357 "The findings presented in this study provide a foundation for developing a framework aimed at enhancing computational methods for off-target sequence prediction. While we can perform large-scale data analysis to discover the characteristics of off-target sequences, more experimental data have been produced using GUIDE-seq and other effective methods. It is expected that further analysis of such data will lead to the identification of precise biological mechanisms of off target mutation generation, and if these mechanisms are validated experimentally, this information can be used to improve computational prediction models in the future". The contribution of the present work is not clearly stated.

Fig.4: Because a) is determined by only 2 sequences, and b) by 144 sequences, could the differences between a) and b) potentially disappear obtaining and analysing the same number of sequences for both the enzymes?

Minor points:

283 "Table 2. Recall, precision and F1 score for different mismatch numbers." The authors should specify the method.

355 "To examine the efficacy of the aforementioned method, large-scale comparisons with

two additional methods based on the Euclidean distance (ED) and Spearman's correlation

coefficient (PCC) were performed". I think the authors mean Pearson correlation coefficient, instead of Spearman's.

357 "The findings presented in this study provide a foundation for developing a framework aimed at enhancing computational methods for off-target sequence prediction. While we can perform large-scale data analysis to discover the characteristics of off-target sequences, more experimental data have been produced using GUIDE-seq and other effective methods. It is expected that further analysis of such data will lead to the identification of precise biological mechanisms of off target mutation generation, and if these mechanisms are validated experimentally, this information can be used to improve computational prediction models in the future". The part: "While we can perform.... more experimental data have been produced" is unclear.

Reviewer #2: I am limiting myself to the three major issues I have raised before:

@a: Regarding the suggested extension of acquiring additional raw data, authors state that "it is not feasible for us to repeat this process".

Regarding software tools: authors explain in their response that "primary aim was to illustrate the current state of available standard programs". They did add CRISTOR in their manuscript, but I miss back to back evaluation results. Furthermore, I still strongly disagree that it suits the claimed aim if authors just present / discuss / analyse two of - in their words - "numerous in silico prediction tools". I would have at least expected a comparison - maybe a table - of strength and weaknesses of these "numerous" tools.

In conclusion, my issue is not appropriately addressed.

@b: I fully agree on the author's assumption: "if two observed patterns deviate in different directions from the random baseline, such differences can be detected using an appropriate statistical method", however, I think they model the starting point of potential deviations incorrectly. It is way more likely to deviate in different directions if you are in the center of a chessboard compared to starting at the top left corner. Random sequences would be the center, the actual sequence pattern is - in my view - much more at the borders.

To be clear, I am not criticizing the metric here, but the assumption of the background model. The fact that other metrics correlate, does not counter my argument, as they would "suffer" from the very same background issue.

I feel we have a misunderstanding here and thus the core of my concern was not addressed.

@c: Looking at the scatter plots of S3 and S4, I feel the results are not very convincing. The computed correlation is high for all data, however, individual data points show huge dissimilarities. The fact that these deviations are not even more pronounced is properly due to the underlying data, i.e. I agree that "the extreme cases mentioned by the reviewer may be uncommon". From the reply to my concern, I cannot clearly see if the authors agree with my assessment, that their metric is sequence independent or not.

Leaving out the theoretical considerations, author's have demonstrated that their metric is roughly on par with state of the art alternatives, thus, my open question might be relevant to the reader.

In conclusion, I am not fully satisfied with how the authors addressed my concerns and would like to see further editions on the manuscript.

Reviewer #3: The authors have addressed all my comments, and thus I have no more comments on the revised article.

**Do you want your identity to be public for this peer review?** For information about this choice, including consent withdrawal, please see our Privacy Policy

Reviewer #1: **Yes: ** Serena Rosignoli

Reviewer #2: No

Reviewer #3: No

---

## [Author Response · Author response to Decision Letter 2]

13 Jun 2025

Response to Reviewers

We are truly grateful to the reviewers for re-reviewing our revised manuscript. We have made the necessary revisions and provide a point-by-point response below.

>Reviewer #1: In this revised version of the manuscript, the authors addressed the comments of the reviewers, however some points require further considerations.

>

>Major points:

>23 "We developed a method to assess sequence pattern similarity and diversity between off-target sites. This method is based on a comparison of ordered relative entropy values for aligned target sequences, and it was compared with two other methods on the basis of Euclidean distance and the Pearson correlation coefficient. The three methods demonstrated clear correlations, indicating their validity."

>366 "These results indicate that the three methods could consistently detect similar or divergent sequence patterns for the 238 pairs"

>It is unclear what is the advantage of the new method based on ordered relative entropy.

The proposed method provides statistical significance in measuring the similarity between sequence patterns, which can help researchers more easily identify biologically relevant patterns. This point has been addressed in the abstract and in the discussion.

>130 "In this study, we first assessed the performance of CRISPOR's computational predictions by comparing them with off-target sites identified through the experiment-based GUIDE-seq method. Additionally, since a number of other novel methods have also been developed, CRISPOR’s predictions were further validated using CRISOT, which outperforms other in silico methods [25]"

>192 "Off-target sequences identified by GUIDE-seq and predicted by CRISPOR were utilized for computing CRISOT scores [25]"

>If CRISOT outperforms other in silico methods, including CRISPOR, the prediction analyses should have been done using CRISOT; and also the 'Comparison between GUIDE-seq data and computational predictions" of 3.2 and Fig.1. Instead, CRISOT was used only to score the off-targets in common between CRISPOR and GUIDE-Seq. It is also possible that CRISOT could have found off targets undetected by both the other two methods.

In this section, our primary aim was to highlight general issues in off-target prediction. CRISPOR is widely used in the field; to our knowledge, publications related to the CRISPOR methodology have been cited over ten times more frequently than those for other methods. Therefore, we focused primarily on this method. Furthermore, the predictive performance of CRISOT, regarding both true positives (active sites) and false positives (inactive sites), was already evaluated by its original authors (Fig. 4, Chen et al., 2023), and we would like to avoid redundant data analysis. Nevertheless, we understand the reviewer's concern and have now included additional data on genome-wide prediction (S2 Fig.), which help readers understand the general performance of the CRISOT method.

>357 "The findings presented in this study provide a foundation for developing a framework aimed at enhancing computational methods for off-target sequence prediction. While we can perform large-scale data analysis to discover the characteristics of off-target sequences, more experimental data have been produced using GUIDE-seq and other effective methods. It is expected that further analysis of such data will lead to the identification of precise biological mechanisms of off target mutation generation, and if these mechanisms are validated experimentally, this information can be used to improve computational prediction models in the future".

>The contribution of the present work is not clearly stated.

To emphasize the significance of our study, we have revised the relevant section accordingly.

>Fig.4: Because a) is determined by only 2 sequences, and b) by 144 sequences, could the differences between a) and b) potentially disappear obtaining and analysing the same number of sequences for both the enzymes?

This is an important point. The number of detected off-target sequences varied significantly in some cases, as previously reported (Fu et al., 2013; Zhou et al., 2022). However, such differences can themselves be considered reflective of underlying sequence pattern divergence. This issue is now addressed and discussed in 4.1.

>Minor points:

>

>283 "Table 2. Recall, precision and F1 score for different mismatch numbers." The authors should specify the method.

We have revised the sentence as follows:

Table 2. Recall, precision and F1 score for different mismatch numbers. True positives, false negatives, and false positives were defined on the basis of the results from GUIDE-seq and CRISPOR.

>355 "To examine the efficacy of the aforementioned method, large-scale comparisons with two additional methods based on the Euclidean distance (ED) and Spearman's correlation coefficient (PCC) were performed".

>I think the authors mean Pearson correlation coefficient, instead of Spearman's.

Thank you for pointing this out. The typo has been corrected.

>357 "The findings presented in this study provide a foundation for developing a framework aimed at enhancing computational methods for off-target sequence prediction. While we can perform large-scale data analysis to discover the characteristics of off-target sequences, more experimental data have been produced using GUIDE-seq and other effective methods. It is expected that further analysis of such data will lead to the identification of precise biological mechanisms of off target mutation generation, and if these mechanisms are validated experimentally, this information can be used to improve computational prediction models in the future".

>The part: "While we can perform.... more experimental data have been produced" is unclear.

The indicated sentence has been revised for clarity.

>Reviewer #2: I am limiting myself to the three major issues I have raised before:

>

>@a: Regarding the suggested extension of acquiring additional raw data, authors state that "it is not feasible for us to repeat this process".

As mentioned in our previous response, in addition to the practical challenges of restarting the entire analyses from data collection stage, the amount of the data used in this study is sufficiently large and diverse to support our conclusion. While using the most up-to-date datasets is ideal, we hope it is understandable that substantial data analysis and related tasks require time, and as a result, the dataset used may not always reflect the latest available data.

>Regarding software tools: authors explain in their response that "primary aim was to illustrate the current state of available standard programs". They did add CRISTOR in their manuscript, but I miss back to back evaluation results. Furthermore, I still strongly disagree that it suits the claimed aim if authors just present / discuss / analyse two of - in their words - "numerous in silico prediction tools". I would have at least expected a comparison - maybe a table - of strength and weaknesses of these "numerous" tools.

>In conclusion, my issue is not appropriately addressed.

As indicated in the title, our primary goal was to elucidate the general characteristics of off-target mutations. We also believe that a deeper understanding of these features will contribute to improving prediction tools. While we appreciate the reviewer's suggestion, presenting a comprehensive evaluation of all currently available tools would, in our view, shift the focus of the paper and potentially mislead readers regarding its core objective.

>@b: I fully agree on the author's assumption: "if two observed patterns deviate in different directions from the random baseline, such differences can be detected using an appropriate statistical method", however, I think they model the starting point of potential deviations incorrectly. It is way more likely to deviate in different directions if you are in the center of a chessboard compared to starting at the top left corner. Random sequences would be the center, the actual sequence pattern is - in my view - much more at the borders.

>To be clear, I am not criticizing the metric here, but the assumption of the background model. The fact that other metrics correlate, does not counter my argument, as they would "suffer" from the very same background issue.

>I feel we have a misunderstanding here and thus the core of my concern was not addressed.

Our focus here is on differences between sequence patterns, and the probability of distinct patterns producing highly similar relative entropy orders is negligibly small. While we acknowledge the concern, it is difficult for us to conceptualize how three independent metrics could all be affected by the same background issue, as the nature of the issue itself remains unclear.

We believe the observed consistency among the three metrics provides a strong indication of the reliability of our results. Nevertheless, to support further investigation and transparency, we have made all WebLogo data publicly available at https://github.com/taitoh1970/ge_off_targets, as noted in 2.4.

>@c: Looking at the scatter plots of S3 and S4, I feel the results are not very convincing. The computed correlation is high for all data, however, individual data points show huge dissimilarities. The fact that these deviations are not even more pronounced is properly due to the underlying data, i.e. I agree that "the extreme cases mentioned by the reviewer may be uncommon". From the reply to my concern, I cannot clearly see if the authors agree with my assessment, that their metric is sequence independent or not.

>Leaving out the theoretical considerations, author's have demonstrated that their metric is roughly on par with state of the art alternatives, thus, my open question might be relevant to the reader.

>

>In conclusion, I am not fully satisfied with how the authors addressed my concerns and would like to see further editions on the manuscript.

First, we acknowledge that the proposed method assesses sequence patterns indirectly through relative entropy values, and we have now clarified this point in 4.1. Second, we recognize the reviewer's concern regarding the scatter plots, when Euclidean distances are large and correlation coefficients are small. Therefore, we introduced a conservative categorization of data points as "semi-divergent" when either the SRCC or the ED alone was insufficient to indicate strong similarity. This approach ensures that only patterns with consistent evidence across multiple metrics are classified as "similar," which we believe strengthens the reliability of the analysis. On the basis of this conservative criterion, we consider the results to be reasonably robust.

Reviewer #3: The authors have addressed all my comments, and thus I have no more comments on the revised article.

We sincerely appreciate the reviewer's thoughtful evaluation and are grateful for the acceptance of the revised manuscript.

---

## [Decision Letter · Decision Letter 2]

Off-target sequence variations driven by the intrinsic properties of the Cas–sgRNA–DNA complex in genome editing

PONE-D-24-52272R2

Dear Dr. Itoh,

We’re pleased to inform you that your manuscript has been judged scientifically suitable for publication and will be formally accepted for publication once it meets all outstanding technical requirements.

Kind regards,

Hodaka Fujii, M.D., Ph.D.

Academic Editor

PLOS ONE

Additional Editor Comments (optional):

Reviewers' comments:

Reviewer's Responses to Questions

**Comments to the Author**

Reviewer #1: All comments have been addressed

Reviewer #2: All comments have been addressed

2. Is the manuscript technically sound, and do the data support the conclusions?

Reviewer #1: Yes

Reviewer #2: (No Response)

3. Has the statistical analysis been performed appropriately and rigorously?

Reviewer #1: Yes

Reviewer #2: Yes

4. Have the authors made all data underlying the findings in their manuscript fully available?

Reviewer #1: Yes

Reviewer #2: Yes

5. Is the manuscript presented in an intelligible fashion and written in standard English?

Reviewer #1: Yes

Reviewer #2: Yes

Reviewer #1: (No Response)

Reviewer #2: All my issues have been addressed.

**Do you want your identity to be public for this peer review?** For information about this choice, including consent withdrawal, please see our Privacy Policy

Reviewer #1: **Yes: ** Serena Rosignoli

Reviewer #2: No

---

## [Editor Report · Acceptance letter]

PONE-D-24-52272R2

PLOS ONE

Dear Dr. Itoh,

I'm pleased to inform you that your manuscript has been deemed suitable for publication in PLOS ONE. Congratulations! Your manuscript is now being handed over to our production team.

Kind regards,

on behalf of

Dr. Hodaka Fujii

Academic Editor

PLOS ONE